# The Botulinum Treatment of Neurogenic Detrusor Overactivity: The Double-Face of the Neurotoxin

**DOI:** 10.3390/toxins11110614

**Published:** 2019-10-24

**Authors:** Chiara Traini, Maria Giuliana Vannucchi

**Affiliations:** Department of Experimental and Clinical Medicine, Research Unit of Histology and Embryology, University of Florence, 50139 Florence, Italy

**Keywords:** urinary bladder, spinal cord lesion, upper lamina propria, detrusor, urothelium, innervation, sensory systems, afferent nerve endings, nerve sprouting

## Abstract

Botulinum neurotoxin (BoNT) can counteract the highly frequent involuntary muscle contractions and the uncontrolled micturition events that characterize the neurogenic detrusor overactivity (NDO) due to supra-sacral spinal cord lesions. The ability of the toxin to block the neurotransmitter vesicular release causes the reduction of contractions and improves the compliance of the muscle and the bladder filling. BoNT is the second-choice treatment for NDO once the anti-muscarinic drugs have lost their effects. However, the toxin shows a time-dependent efficacy reduction up to a complete loss of activity. The cellular mechanisms responsible for BoNT effects exhaustion are not yet completely defined. Similarly, also the sites of its action are still under identification. A growing amount of data suggest that BoNT, beyond the effects on the efferent terminals, would act on the sensory system recently described in the bladder mucosa. The specimens from NDO patients no longer responding to BoNT treatment displayed a significant increase of the afferent terminals, likely excitatory, and signs of a chronic neurogenic inflammation in the mucosa. In summary, beyond the undoubted benefits in ameliorating the NDO symptomatology, BoNT treatment might bring to alterations in the bladder sensory system able to shorten its own effectiveness.

## 1. Introduction

Botulinum neurotoxin (BoNT) represents one of the most powerful and versatile drugs currently available in the pharmaceutical market. It is effective and safe in the treatment of several disorders spanning among numerous medical specialties such as ophthalmology, orthopedy, dermatology, urology, pain medicine and plastic surgery [1,2,3,4,5,6]. Remarkably, BoNT also has important central effects that are not solely considered “side-effects” but rather additional therapeutic actions of the drug [7].

BoNT is also used in the neuro-urological field for the treatment of the neurogenic detrusor overactivity (NDO), a bladder dysfunction due to supra-sacral spinal cord injuries of different origins and whose main symptoms are urinary frequency, urgency, and incontinence [8]. The DIGNITY (double-blind investigation of purified neurotoxin complex in NDO) research program [9,10] and the subsequent prospective study [11], showing significant improvement in the urodynamic parameters and in patient quality of life after BoNT subtype A (BoNT/A) intra-vesical injections, validated the success of the toxin in NDO therapy. Indeed, the access to BoNT/A treatment has given to NDO patients an additional chance to control the symptomatology when the therapies with anti-muscarinic acetylcholine receptor (mAChR) or α-adrenergic receptor (-adrenoR) antagonists reduce or lose their effectiveness. However, as in the other fields of application, the BoNT/A treatment needs to be periodically repeated in NDO. This necessity could depend on the compensatory nerve sprouting employed by the neurons to overcome the block of synaptic vesicular release caused by the toxin [12,13]. Nevertheless, the BoNT/A efficacy of each administration decreases with time and, after a variable number of years [14], the toxin loses its therapeutic effects. 

Finally, regarding the BoNT site(s) of action in the bladder, an increasing amount of data from human specimens and animal models confirms the toxin’s ability to block the efferent nerve terminals in the detrusor but also indicates that BoNT/A affects the sensory signaling generated in the mucosa. 

The present review aims to describe the BoNT positive effects in NDO and the possible sites of its action. Moreover, emphasis will be placed on the changes observed in the sensory system in bladder specimens of NDO patients in which BoNT was no longer effective. 

## 2. Bladder Physiology

### 2.1. Bladder Innervation

The bladder physiology results from the interplay between the central and the autonomic nervous system [15]. The parasympathetic and sympathetic pathways, that regulate the bladder’s basic physiology, undergo to the cortical motivational system which projects to spinal motor neurons. 

Briefly, the involved autonomic nervous centers are:(1)The sympathetic thoracolumbar center (T10–L2) which, by mean the hypogastric nerve, sustains the bladder muscle relaxation (bladder compliance) and the internal sphincter contraction allowing the accommodation of the growing urine volume without increasing intra vesical pressure;(2)The parasympathetic sacral center (S2–S4) which, by means the pelvic nerve, sustains the detrusor contraction and the internal sphincter relaxation, creating the conditions for bladder emptying.

Since the micturition is a voluntary action, it involves different supra-spinal areas which project first to pontine micturition center and then to the sacral Onuf’s nucleus. From this nucleus the pudendal nerve ensures the voluntary external urethral sphincter relaxation and urine voiding (Figure 1A).

### 2.2. Sensory System

The perception of the bladder state is ensured by an integrated system encompassing the urothelium, the underlying lamina propria (LP), namely the connective tissue between the transition epithelium and the detrusor, and the local afferent terminals (Figure 2A) thus including the entire mucosa [16,17,18,19,20]. 

#### 2.2.1. The Urothelium

The urothelium has “*neuronal-like properties*” [21] through which it takes part in the bladder’s sensory system [21]. Application of physical or chemical stressors *in vitro* preparations entails the release of several molecules from the urothelium (adenosine triphosphate, ATP, nitric oxide, NO, ACh, prostaglandins, PGs), which have autocrine and paracrine actions [19]. 

The ATP is released through non-vesicular and vesicular mechanisms. In rodent bladder, the application of selective inhibitors of pannexin and connexin hemichannels significantly reduced the non-vesicular ATP release stimulated by the instillation of bacterial lipopolysaccharides or mechanical distention [22,23], and immunofluorescence showed the presence of these mechanosensitive channels in the urothelium [22]. The existence of an ATP vesicular release was mainly supported by the inhibitory effect of BoNT/A observed in rodents [24,25,26]. 

Using reverse transcription-PCR it has been demonstrated that rat urothelium expresses NO synthases [27] and, upon chemical stimulation, produces NO [27,28]. 

The human urinary bladder mucosa produces different types of eicosanoids, many of which are PGs [29]. In mouse urothelium, it was shown that the stretch-induced PGs release enhances ATP release [30] and, in the guinea pig mucosa, the ATP increased PGs release [31], creating a positive feed-back loop [19].

Finally, the urothelium, similarly to other non-neuronal cells, releases ACh [32] and this release implies a non-vesicular mechanism. In fact, it has been reported that the rat urothelium lacks vesicular ACh transporter VAChT [32] and, in guinea pig urothelial cells, the application of brefeldin (which disrupts vesicular exocytosis) and of BoNT/A did not affect ACh release [32,33]. Unlike ATP, the non-vesicular ACh release does not employ the connexin/pannexin hemichannels, but rather the cystic fibrosis transmembrane conductance regulator (CFTR) channels as their block reduced the ACh release from guinea pig urothelial strips [33]. It has also been postulated that ACh crosses the CFTR channels either alone or bound to a cofactor [33]. 

#### 2.2.2. The Lamina Propria

The LP has been defined “*the communication center*” of the sensory system playing an integrative role between the urothelium signals and the local afferent terminals [19,34]. LP holds two peculiar cell types: the myofibroblast (Myo) and the telocytes (Tc) which form a 3D network making numerous cellular junctions (Figure 3A) [20,35,36,37]. This network would perform structural and sensory functions: as a scaffold, the network supports the organ during volume changes because of filling and emptying processes, as a “*stretch receptor*” [35] it responds to the distention grade of its own meshes. Further, since Myo and Tc express vanilloid, purine, and muscarinic receptors, they could recognize the urothelium signaling and mediate/potentiate/propagate the information from the urothelium to the nerve afferents, working as a “*functional syncytium*” [19]. Finally, the LP releases inflammatory mediators such as histamine, leukotrienes, PGs; [38] while it is not proved whether its cells release neurotransmitters. In summary, a highly integrated information would develop in the LP and, through the excitation of afferent fibers, be transmitted to the related spinal cord areas. Additionally, the Myo/Tc 3D network plays also a role in the spontaneous contractile activity of the detrusor, probably aimed at maintaining the shape of the organ during emptying [17,19,20]. The source of this activity is still not defined; however, the “*urotheliogenic hypothesis*” rather than the myogenic one (intrinsic contractile activity of smooth muscle cells) is more consistent with the experimental data [19]. The stronger results supporting such a hypothesis are the mechanical contractions recorded from bladder mucosa strips free of the muscle component and ascribed to both the discontinuous muscularis mucosa and the Myo [17,19]. This activity might be driven by the ATP released from urothelium and propagated up to the detrusor through the Myo/Tc network [17,19].

## 3. Neurogenic Detrusor Overactivity (NDO)

### 3.1. Pathophysiology

Lesions due to traumatic injuries or neurologic diseases, which affect the aforementioned neuronal pathways at any level, lead to lower urinary tract dysfunctions which are overall defined as “*neurogenic bladder*” [39]. The location of the lesion is mildly predictive of the type of dysfunction because of the wide range of pathologies involved and symptoms declared by the patients. As a general criterion, a supra sacral spinal cord lesion (Figure 1B) (common in traumatic spinal cord injury, demyelinating diseases or spinal cord cancers) or supra pontine or cerebral cortex damages (occurring after stroke, in multiple sclerosis, Parkinson disease or brain cancers) lead to the development of NDO [8]. The main NDO signs are highly frequent involuntary detrusor contractions and uncontrolled micturition events for small urine volume. NDO is often associated with detrusor-sphincter dyssynergia (namely, the loss of coordination between bladder contraction and internal urethral sphincter relaxation). The several urodynamic tests available (cystometry, uroflowmetry, etc.,) allow the diagnosis and give the possibility to follow the disease progression over the time [8]. The loss of voluntary micturition control is one of the most disabling aspects of NDO and it represents the major reason of social isolation since patients prefer abandoning the rehabilitation process, medical and social appointments rather than being embarrassed by their bladder dysfunction [40,41]. Therefore, regaining the management of urine voiding is one of the highest priority of patients’ rehabilitation.

The pathogenesis of NDO is still unclear; two different mechanisms have been proposed to explain the symptoms: 1. An increased afferent signaling; 2. an abnormal handling of efferent activity. The involvement of the urothelium [21] and of the sensory cellular network of the LP in NDO pathogenesis finds continuous confirmation. However, the distinction between sensory or motor causes is simplistic and the strict correlation between the systems implies the much larger concept of “*sensorimotor neuromodulation”* [19].

### 3.2. First-Choice Therapy

The therapeutic strategies for NDO are aimed at preventing the urine reflux and renal damage resulting from high intra-vesical pressures and incontinence. The first-choice drugs are the mAChR and/or α-adrenoR antagonists, orally or intra-vesical administered, coupled to assisted bladder drainage (intermittent or suprapubic catheterization) [42]. The rationale of the mAChR antagonists treatment resides in the observation of an increase in muscarinic receptor density and sensitivity in NDO patient [43]; besides, several clinical trials have demonstrated that mAChR antagonists decrease detrusor pressure, improve bladder capacity, and ameliorate the quality of life of these patients [44]. Unfortunately, the oral mAChR antagonists therapy causes adverse effects in 61% of the patients and, in any case, the effectiveness of these drugs is reduced over time and the increase in dosage often raises up or worsens the side effects [43]. When the mAChR antagonists permanently lose their efficacy, clinicians recommend switching to intra-bladder BoNT/A injections before choosing surgery [39].

### 3.3. Second-Choice Therapy: BoNT/A Injections

In NDO patients, BoNT/A is administered under partial or general anesthesia through a series of injections into the detrusor using a rigid or flexible cystoscopy excluding the dome to prevent the erroneous spill in the peritoneal area [45,46]. The optimal doses are 200U up to 300U since no further improvement has been recording with higher doses [9,10]. The efficacy of BoNT/A is monitored by the results of urodynamic parameters (detrusor compliance, bladder capacity, maximal detrusor pressure) and by patients’ opinion on their wellness [45,46]. The effectiveness of a BoNT/A injection lasts, on average, for 9 months although 26.0% of the patients experience beneficial effects up to 12 months or longer during the first 4 years of cyclic treatments [47,48]. Over time, the BoNT/A effectiveness decreases until a complete loss of its therapeutic effects. On average, 12–14 years since the first injection, 60% of the patients still present beneficial effects while 40% of them have discontinued the therapy [14].

## 4. BoNT Sites of Action in the Bladder

A growing amount of data supports the assumption that the BoNT sites of action in the bladder are more numerous than postulated (Figure 4) and reflect the complexity of the organ (patho) physiology. The demonstration that the toxin, once injected in the detrusor, spreads from the muscle into the suburothelium is one of the most important observations to sustain this assumption [49]. 

In this way, the toxin has the chance to reach neuronal and non-neuronal cells residing in the deepest layers and expressing the functional targets of BoNT such as the soluble *N*-ethylmaleimide soluble fusion protein (SNARE) and the synaptic vesicle protein 2 (SV2) SV2 (see Section 4.3). 

The toxin effects depend on its uptake by the cells. In nerves, the BoNT/A uptake changes with the frequency of firing and with the amount of SV2 expressing nerve terminals. Notably, in the mouse bladder, the parasympathetic nerves express the highest percentage of SV2 (95%) but their activation is restricted to the fast contraction phase, during voiding function. The sympathetic nerves and the afferent terminals express a lower percentage of SV2 (58% and 69%, respectively) [50] but have a long period of firing, lasting as much as the filling phase [51]. These results support the conclusion that BoNT/A can be taken up by efferent and afferent nerve terminals and that the long-lasting inhibition of contraction is ascribable to a parallel effect on both. 

### 4.1. Efferent Nerve Terminals

The application of BoNT in the bladder temporally blocks the release of ACh from the efferent terminals. The reversibility of the block has been assessed using a radiochemical method in detrusor muscle strips from BoNT/A-treated rats, where the release of ACh was significantly inhibited at higher frequencies of electrical field stimulation (able to mimic the effect of the parasympathetic nerve over activity) 5 days after the injection and it recovered to control values at the 30th day [52]. A further confirmation comes from experiments in spinal cord transected mice where the electrical stimulation of spinal nerves abolished the muscle activity in BoNT/A-injected bladders [34]. Since a cholinergic and purinergic co-transmission is present in the bladder [53], BoNT/A should affect the release of both these excitatory neurotransmitters. 

### 4.2. Afferent Nerve Terminals

It has been demonstrated that in spinal cord-transected mice, BoNT/A injection reduces the afferent activity evoked by mechanical (stretch) and chemical (capsaicin) stimulation [34] and it has been assumed that the increased firing of the afferents caused by the lesion favors a high uptake of the toxin into these terminals [34]. In particular, the toxin would be able to prevent the substance P release [34,54] and the following autocrine stimulation of the neurokinin 1 and 2 (NK1 and NK2, respectively) receptors [18,55]. This would explain also the efficacy of BoNT/A injection in the bladder pain syndromes [54,56,57]. Moreover, since the expression of the capsaicin receptor transient receptor potential vanilloid 1 (TRPV1) on the cell surface also depends on a SNARE-mediated exocytosis [58] and since the overexpression of TRPV1 observed in the afferent terminals of NDO bladders is reduced after BoNT/A treatment [59], it is reasonable to assume that the toxin also leads to a normalization of the receptor trafficking, as further therapeutic effect [60]. A similar effect might occur for the purinergic excitatory P2X3 receptor expression on the afferents. These receptors are activated by the distension-mediated ATP release from urothelium and are strictly connected with pain signaling transduction [60,61]. In NDO patient, P2X3 receptors are overexpressed and after few BoNT/A injections their expression returns to control value [59]. 

### 4.3. Urothelium 

As previous reported, the bladder sensory system, beyond the afferent terminals, includes the urothelium and the Myo/Tc 3D network. The ability of the urothelium to release several transmitters is well-known [19]. However, the question whether and how BoNT blocks this transmission is still open. It has been demonstrated that human urothelium has a BoNT-insensitive ACh release [32,62] and a BoNT/A-sensitive ATP release [24]. This latter has been observed in rat and human urothelial cell cultures and has been correlated with the expression of SV2 and of SNAP-23/-25 identified by immunoblot technique [24]. Similar results were obtained in a mouse bladder in vitro preparation that measures the afferent nerve activity and the intravesical pressure [63]. These data found confirmation in an animal model of NDO. After spinal cord lesion, the release of ATP in the urothelium is consistently higher with respect to control and it is significantly inhibited by the BoNT/A application [64]. All these results indicate the urothelium (and specifically the purinergic transmission) as a second site of BoNT/A action within the bladder sensory system. It needs to mention that, using immune labeling, no SV2 and SNAP-25 expression was found in human urothelium from cadaveric organ donors [50]. 

### 4.4. Lamina Propria

The Myo and Tc, connected by cell to cell junctions, build a 3D network that forms the organ scaffold and behaves as an integrating stretch-receptor. The 3D structure, following organ distension and relaxation, contributes to the compliance of the bladder and avoids anomalous organ deformation [20,65]. At the same time, the expression of cGMP, the target molecule of NO [66] and of NK1 [61], vanilloid [67], muscarinic [68] and purinergic (both metabotropic P2Y and ionotropic P2X [69,70]) receptors on the Myo, makes the network able to respond to signaling from urothelium and afferent terminals. The activation of these receptors generates Ca^2+^ waves that spread within the network up to the detrusor muscle [71]. Among the connective cells, a vesicular release of transmitter from Tc, although postulated in other organs [72], has not been demonstrated in the bladder. 

## 5. The Double-Face of the Neurotoxin

### 5.1. Cyclic BoNT Treatments Generate Substantial Changes in the Bladder Wall

The progressive loss of the BoNT efficacy is typical of the toxin wherever it is injected. As reported in skeletal muscle, the block of the nerve terminals causes the sprouting of new nerve endings as the attempt to restore the synaptic activity. In some cases, the development of antibodies against the toxin is responsible for the definitive loss of efficacy [73]. In the bladder, however, the picture shows several peculiarities. First, during the period when BoNT/A is effective most of the patients complain cyclical lower urinary tract (LUT) infections as the only common adverse event [47,48] and statistical evaluations have shown a significantly higher rate of LUT infections in BoNT/A-treated patients compared to those receiving the placebo [74]. Second, the bladders of NDO patients no longer responding to BoNT/A and subjected to cystoplasty display several signs of aging (in spite of their young age) such as inflexibility and fibrosis, reduction in luminal volume and organ dimensions (personal observation) and an increase of the thickness [75]. These findings are likely the result of structural changes that proceed over time and whose causes are multiple. The repeated BoNT/A treatments cause micro-wounds in the detrusor followed by micro-scaring able to reduce the bladder compliance and favor inflammatory events [76]. The classical template for each treatment, in fact, consists of at least 20 injections distributed in the lateral wall [39]. Furthermore, social difficulties, or the erroneous time assessment of bladder filling (depending on the water and food management, physical activity, etc.,) and the absence of perception can prolong the filling phase and cause the excessive distension of the organ wall. The subsequent, immediate emptying through auto-catheterization stimulates the bladder wall in the opposite direction. Over time, the recurring exceeding the physiological limits affects the *elastic* capability of the bladder and reduces the organ compliance. Objectively, the possibility to acquire information on the tissue/cellular changes due to the disease and/or to the toxin is limited by the difficulty to obtain human samples at the advanced stages of the therapies and by the inability of generating these stages in the animal models. Literature reports important differences in the bladder responses after the first BoNT/A injection or at the end of the treatment cycles. Biopsies of the urothelium and LP from NDO patients before and after the first BoNT/A injection showed a widespread axonal degeneration, a slight axonal sprouting, Schwann cell activation [76] and absence of fibrosis [75]. However, it has also been observed an increase of the excitatory nerve terminals in the LP of NDO patients before the BoNT/A treatment that was restored to control values after one or two toxin injections [59]. Finally, almost half of the patients showed a chronic inflammatory infiltrate that was extended up to 80% of the patients after the first treatment [77]. 

On the contrary, in full thickness bladder specimens from patients no longer responsive to BoNT/A, it was reported an overt chronic inflammatory picture (Figure 3B,C), a consistent fibrosis of the detrusor, a patchy denervation of the organ [37,78,79], and a significant increase in the nerve terminals, likely excitatory, in the LP, precisely in the upper portion (ULP), together with a significant increase of pro-inflammatory molecule expression in the glia cells (Figure 5) [78].

The bladder denervation has been interpreted as a BoNT/A independent phenomenon. It would be the direct consequence of the spinal cord injury with the lack of the central nervous system trophic action and consequent neuronal loss at the peripheral ganglia [78,79]. The nerve damage would be also implicated in the generation of fibrosis in the detrusor [78,79].

At variance, the increase in the ULP innervation, interpreted as a sprouting of the afferent terminals, was ascribed to BoNT/A treatment and considered the proof that the afferent terminals are a target of the toxin [78]. If so, the NDO symptomatology evolution and the loss of the toxin effectiveness, might found reasonable explanations. In particular, the excess of the nerve terminals likely lowers the threshold of excitation; moreover, since this sprouting was untidy, the communication/integration of signaling inside the LP sensory system might be significantly altered; finally, since these new afferent terminals are likely excitatory [78], they contribute to the generation and maintenance of the chronic neurogenic inflammation [77,80].

### 5.2. Sprouting and Chronic Neurogenic Inflammation

The correct signaling among the components of the urinary bladder sensory system depends on a suitable distance between the cells. In NDO two events hit this system: i) The untidy distribution of the newly formed nerve terminals; ii) the chronic neurogenic inflammation constantly present in NDO patients and primarily localized in the LP (Figure 2B). The inflammation stretches and stresses the Myo/Tc network extending the meshes and increasing the distances among the cells [37,78]. 

The inflammation, present from the early stages of the disease, acquires the features of chronicity at advanced stages, being rich in lymphocytes and plasma cells. The immune cells are diffuse in an edematous extracellular matrix or grouped into granulomatous structures (Figure 2B,C) [37,75,77,78]. This condition, primarily generated by irritative events (see above), over time becomes neurogenic because of the contribution of the local nervous components. The neuropeptides and the neuromodulators (substance P, calcitonin gene related protein, neuropeptide Y, ATP) are key elements for the establishment of neurogenic inflammation and they are released by activated sensory nerve endings of the LP [80,81,82]. These molecules can mediate different and multidirectional interactions in multicellular networks, acting on endothelial cells, immune cells, smooth muscle cell, and nerve terminals [83]. These effects produce a positive feedback loop which could turn into a vicious circle and lead to a chronic neurogenic inflammation in determined conditions [84]. In NDO bladder, it is conceivable that BoNT/A turns the virtuous circle into a vicious one, because of the afferent sprouting and contributes to amplify and maintain the neurogenic inflammation over time.

## 6. Conclusions

BoNT/A is a great resource for NDO patients. It has significantly extended the time of pharmacological treatment before surgery. Further, being locally injected, does not enter the systemic circulation and results devoid of side effects. It is a second-choice therapy only because of the hospitalization required for its administration. 

Interestingly, BoNT acts on a broader number of targets than expected, mirroring the complexity of the bladder physiopathology. Indeed, by the block of the vesicular release, the toxin acts on afferent and efferent endings, on the urothelium and, possibly, on the Myo and Tc that form the scaffold of the organ. Nevertheless, the exhaustion of the toxin efficacy in NDO is an irreversible fate. 

Based on the literature data, we hypothesize that the BoNT plays a double game in the events that sign its fate since, together with its therapeutic and beneficial effects, it induces local changes responsible for the lack of its effectiveness. The main change is the significant nerve sprouting of excitatory afferent terminals. This sprouting from one side confirms the ability of the toxin to act on afferent endings; from the other side, this response supports and feeds the local inflammation that acquires neurogenic features over the time. 

In this view, a pharmacological approach limiting neurogenic inflammation (eventually coupled to BoNT/A injections) could be effective in slowing the bladder deterioration and, likely, in lengthening the toxin treatment effectiveness.

## Figures and Tables

**Figure 1 toxins-11-00614-f001:**
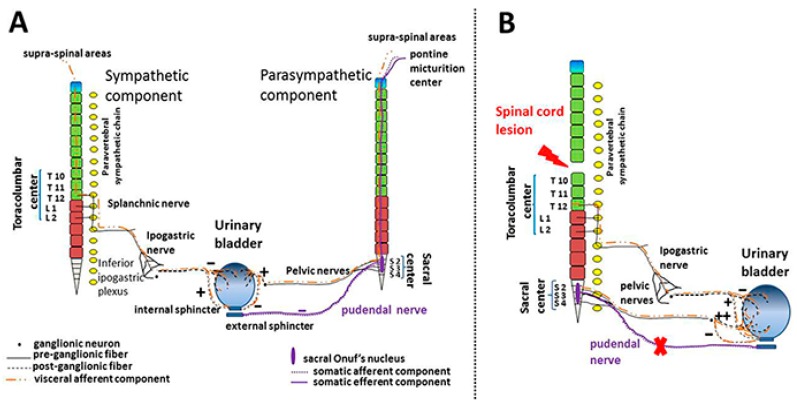
Schematic drawing of the nerve pathways that regulate the urinary bladder physiology in healthy (**A**) and NDO (**B**) patients. (**A**) The sympathetic thoracolumbar center (left side) (T10–L2) mediates the relaxation (−) of the detrusor and the contraction (+) of the internal sphincter of the bladder through one of the branches of the hypogastric nerve that originate in the lower homonymous plexus. The parasympathetic sacral center (right side) (S2–S4) sustains the detrusor contraction (+) and the internal sphincter relaxation (−) by mean of the pelvic nerve. The hypogastric and pelvic nerves have an afferent component that projects the visceral sensitivity to the spinal cord. The conscious perception of the filling and the voluntary action of micturition involve supra-spinal areas. The afferent component through the pudendal nerve projects the perceptions to the sacral Onuf’s nucleus and from this nucleus to the pontine micturition center up to the cerebral cortex. The efferent signaling follows the somatic descending pathways projects to the motor areas of the same nucleus from which efferent fibers, through the pudendal nerve, relax (−) the voluntary external sphincter. (**B**) Supra-sacral lesion abolished the voluntary micturition control mediated by the pudendal nerve (X) and causes parasympathetic overactivity (++) mediated by the pelvic nerve.

**Figure 2 toxins-11-00614-f002:**
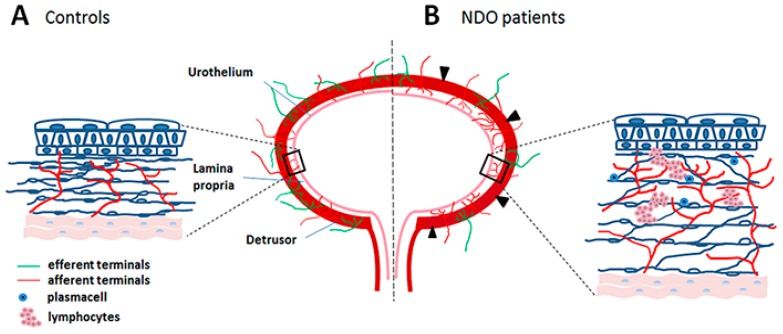
Schematic drawing of the bladder innervation in controls (**A**) and in NDO patients no longer responding to BoNT/A (**B**) In (**A**) the afferent (red lines) and efferent (green lines) innervations are regularly distributed in the detrusor, the lamina propria and the urothelium. In (B) the detrusor and the lamina propria (LP) present a patchy denervation (arrowheads); in the upper LP, a significant increase in the afferent nerve terminals and in the thickness, numerous immune cells and edema, all signs of neurogenic chronic inflammation, are present. The content of the small black squares is enlarged in the lateral drawings to appreciate the differences between controls and NDO patients.

**Figure 3 toxins-11-00614-f003:**
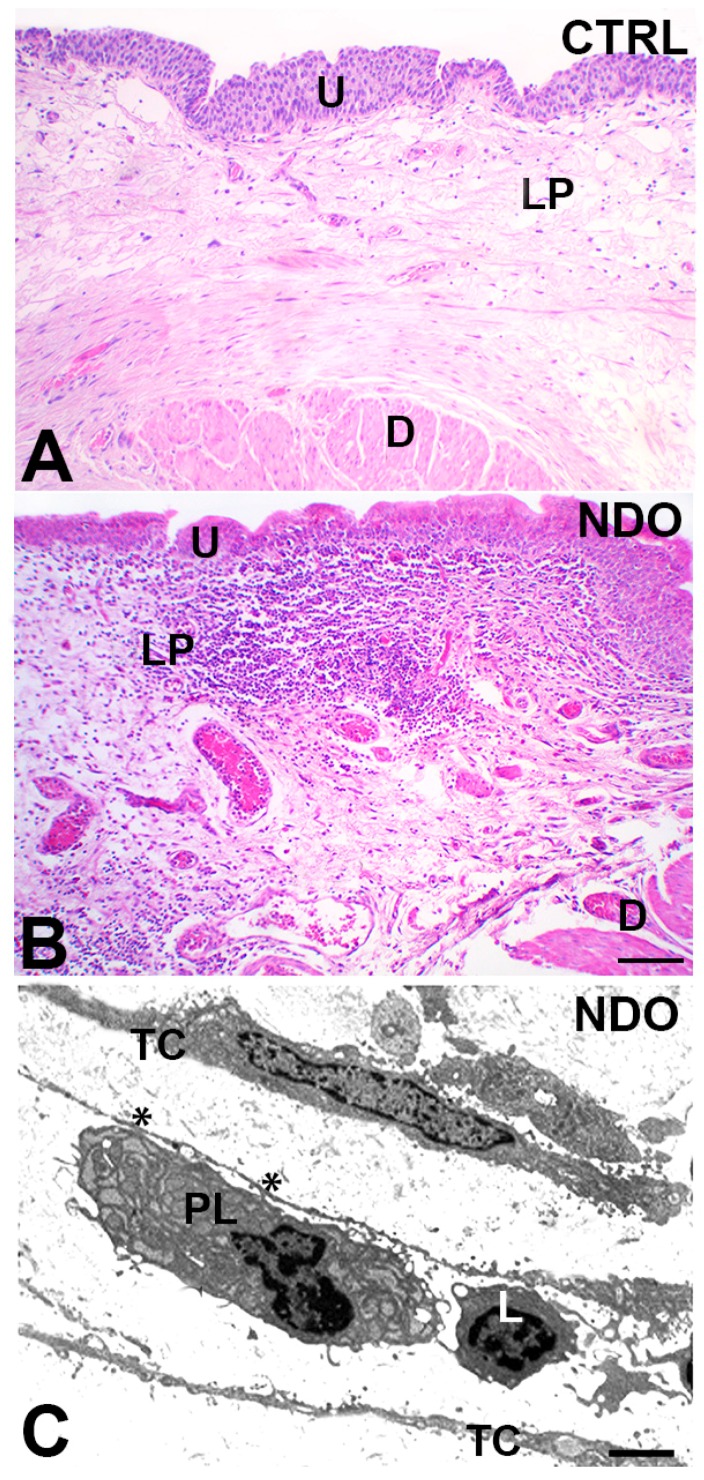
Bladder lamina propria. (**A**,**B**) hematoxylin and eosin staining. (**A**) controls: in the lamina propria several rows of cells made of myofibroblasts and telocytes form a network with narrow meshes. Numerous and small capillaries are present. (**B**) NDO patients no longer responding to BoNT/A: the upper lamina propria is thicker than in controls, rich in immune cells infiltrate organized in large groups and contains dilated capillaries. (**C**) Transmission electron microscopy. NDO patients: cells identifiable as typical telocytes for their small oval body and long and thin processes are near or make contacts (asterisks) with the immune cells. U: urothelium, LP: lamina propria; D: detrusor; TC: telocytes: PC: plasma cell; L: lymphocyte. Calibration bar: (**A**,**B**) = 25 µm; (**C**) = 1.6 µm. The imagine was modified from [37].

**Figure 4 toxins-11-00614-f004:**
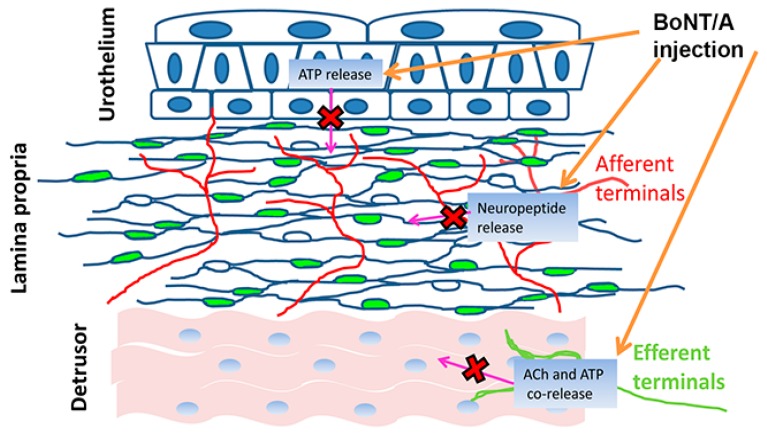
Schematic drawing of the BoNT/A action sites in the bladder wall. After the intra-vesical injections, the toxin spreads from the detrusor to the connective tissue up to the urothelium. The presence of the toxin target proteins (SV2, SNARE) leads to the block of the vesicular release of neurotransmitters, neuropeptides, and neuromodulators from the efferent and afferent terminals and from the urothelium.

**Figure 5 toxins-11-00614-f005:**
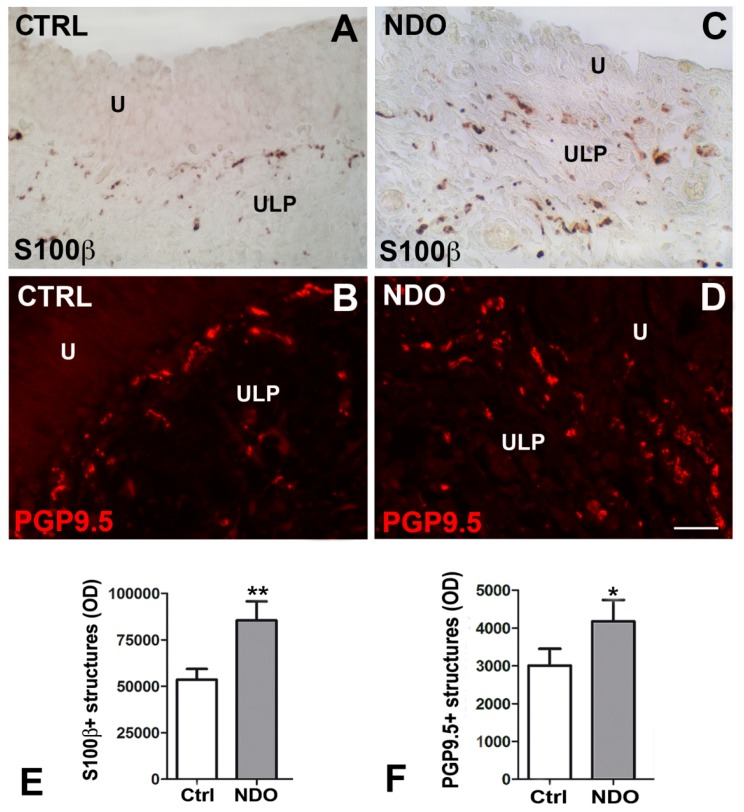
Upper lamina propria. Glial cells S100β (**A**,**C**) and pan neuronal marker PGP9.5 (**B**,**D**) immunohistochemistry in controls and NDO patients no longer responding to BoNT/A. In the upper portion of the LP, the IR shows thin varicose nerve fibers, which in controls (**A**,**B**) are mostly localized under the urothelium, while in the NDO patients (**C**,**D**) are more numerous and dispersed in the entire thickness of the upper LP. The quantitative analysis (**E**,**F**) of the two labeling shows a significant increase of the S100β- and PGP9.5-IR in the NDO patients with respect to the controls. U: urothelium; ULP: upper lamina propria; OD: optical density. Calibration bar: (**A**–**D**) = 25 μm. ** *p* < 0.01; * *p* < 0.05. The imagine was modified from Traini et al. [78].

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
