# Peer review of "The Botulinum Treatment of Neurogenic Detrusor Overactivity: The Double-Face of the Neurotoxin"

_toxins, 2019, doi:10.3390/toxins11110614_

Round 1
Reviewer 1 Report
This paper seemed to be revised successfully.
Author Response
We thank the reviewer for the positive comments.
A further re-revision by a native English speaker has been done.

Reviewer 2 Report
The revised version of this article seems to me much more satisfactory, since substantial modifications have been made and improve the whole.
Some minor changes must be taken into account:
L22: BoNTs act or BoNT/A acts L47: validated the success L52: α-adrenoreceptor L126 : I think it should be “non-vesicular and vesicular mechanisms” (as in L130). Please correct if necessary. L129: the presence L145: Unlike ATP, L146: the connexin/pannexin hemichannels, but rather L182: Hematoxylin and eosin staining L189 and 419 : The image. Make sure that a copyright does not apply for image reproduction (see through the editorial process) L225: that mAChR antagonists L275: soluble fusion protein (SNARE) and the Synaptic Vesicle Protein 2 (SV2) (see section 4.3) L347: Ca2+ L371: BoNT/A-treated patients L371: compared to those receiving the placebo L372-374: Not clear. Rephrase. L464: keys elements L476: turns the virtuous circle into a vicious one, due toAuthor Response
Please, see the attachment.

Reviewer 3 Report
This is a revised version that reports a treatment of neurogenic detrusor overactivity (NDO) with botulinum neurotoxin and data obtained from NDO affected patients. This has been largely revised according to my comments. The authors have added Fig. 1 that serves to understand a neuronal circuitry about bladder physiology and Fig. 2 showing possible bladder innervation in controls and NDO patients. Furthermore, inaccurate explanation about receptors has been revised. There are, however, many of minor points that may be useful to amend this article, as follows:
line 6: not “characterized” but “characterize”? Line 10: “drug” following “anti-muscarinic”? Line 11: not “which” but “what”? Line 32: ”and” following “pain medicine”? Line 43: ACh should be defined in this line but not in line 103; “a” should be put before “-adrenoR”. Line 45: not “to” but “on”? Lines 73, 76 and 83: not “centre” but “center”; not “fibre” but “fiber”. Line 103: not “PG” but “PGs”. Line 107: not “.” but “,”; not “prsence” but “presence”. Line 113: please correct “several PGs”, because this is not a sentence. Line 134: not “prostaglandins” but “PGs” (see line 103). Line 150: please put “NDO patients” in (C). Line 182: “a” should be put before “-adrenoR”. Line 241: please put “, respectively” following “NK2”. Line 272: please put “and” following “P2Y”. Line 335: please delete period.Author Response
We thank the reviewer for the very useful and accurate comments. Attached is the response, point-by-point, to all the minor points.

This manuscript is a resubmission of an earlier submission. The following is a list of the peer review reports and author responses from that submission.
Round 1
Reviewer 1 Report
This is an interesting review article and suggests new indication for BTX treatment. However, there were many typo errors and grammar errors. Though this was the article related to bladder dysfunction, there have been many papers about new BTX applications in other fields. These papers are frequently found in Toxin. Thus, they should be cited, too.
Reviewer 2 Report
toxins-574925
The article entitled "The botulinum treatment of neurogenic detrusor overactivity: the double-face of the neurotoxin” is a review devoted to the therapeutic use of botulinum neurotoxin type A (BoNT/A) in the context of NDO. It proposes a presentation of the organization of the bladder (consisting in 3 concentric parts), the mechanisms at the origin of the pathology and the mode of action of the BoNT/A which effects target both motor and sensory nerve endings.
This review is interesting, but it contains a number of points that significantly reduce its level. First of all, innumerable mistakes in English have to be corrected.
The review proposes that BoNT, which is one of the treatments currently underway for NDO, actually acts on the efferent motor nerve fibers that innervate the detrusor, but also on the afferent sensory fibers that innervate the lamina propria. By acting on them, it would induce deleterious side effects for the patient. The question I ask myself is this: are such side effects observed for the other therapeutic uses of BoNT/A or /B. In skeletal muscles (treatment of dystonia), there are also afferent sensory fibers: are they targeted by BoNT treatment? This point needs to be clarified.
I think that the first section of the review on the physiology of the bladder should be illustrated with a figure showing a normal situation and a pathological situation so as to present clearly the physiological context of the use of BoNT.
Ref. [1] gives a very incomplete overview of the therapeutic use of BoNTs (4 uses are mentioned). It must be added an exhaustive review of the medical applications of BoNTs: Chen, 2012. Clinical Uses of Botulinum Neurotoxins: Current Indications, Limitations and Future Developments. Toxins 4 (10): 913-939. doi: 10.3390/toxins4100913 can be quoted here.
L38-39: I ask the authors not to use the same abbreviation for botulinum neurotoxins (BoNT) and botulinum neurotoxin type A (BoNT/A) to avoid confusion in the reading of their review, even if the BoNT A is the only one used for NDO management. BoNT must therefore be reserved for botulinum toxins in general; BoNT/A for serotype A only.
L82-85 : what about the local release of ACh, NO and prostaglandin ? It should be through a vesicular process. Please precise. L86-87 are not clear on this point.
Minor points
L36 : [3,4]
L41-42 : please precise: muscarinic receptor or alpha-adrenergic receptor antagonists ?
If so, use the abbreviations mAChR and α-adrenoR, since they appear frequently in the text.
It would be better to replace “anti-muscarinic” by “mAChR antagonists”
L45: a ref is missing. Add “Duchen LW, 1972. Motor nerve growth induced by botulinum toxin as a regenerative phenomenon. Proc R Soc Med. 65(2): 196-197. PubMed PMID: 5085043”
L50: the efferent nerve terminals
L51: “in the skeletal muscle” to be replaced by “at the neuromuscular junction”
L59: results
L80: acetylcholine (ACh)
L80: adenosine triphosphate
L85: see section 4
L85: 15,16
L96: since Myo and Tc cells
L97: they would be able to recognize
L111: to the Myo cells. This activity
L139: aimed at preventing
L140: and…and
L142: intermittent or suprapubic catheterization
L163: the patients can undergo cyclic administrations
L164: over time. Is there a mean time after which BoNT is no longer effective? If so, it should be added.
L171-173: add a reference.
L172: Soluble N-ethylmaleimide-sensitive fusion protein Attachment Protein Receptor (SNARE), Synaptic Vesicle Protein 2 (SV2)
L177, L182: SV2
Fig.1: detrusor is written detrusore. The figure caption contains several mistakes that need to be corrected.
L190, L220: ACh
L195: recorded
L207: the capasaicin receptor TRPV1
L214: returns
L283: it was reported
Fig.2 and Fig.3: where has been done this work? Are these original results or reproduced from an article? Please precise in the caption.
Fig.3: the calibration bar is missing in the images
L363: Naturally, given the public health issues and the widespread therapeutic use of BoNTs, any support from a company that has subsidized all or part of this review, in any form whatsoever, must be mentioned.
Reviewer 3 Report
This review article reports a treatment of neurogenic detrusor overactivity (NDO) with botulinum neurotoxin (BoNT) and data obtained from NDO affected patients. Figure 1 gives a schematic diagram of possible targets of BoNT in the bladder wall, while Figures 2 and 3 represent an image of bladder wall and nerve ending distribution in the bladder lamina propria (LP), respectively, in control and NDO affected patients. Figures 2 and 3 are inappropriate in this review, because data shown in these figures appear to be not directly to BoNT treatment. It may be better to give a schematic diagram showing a morphological change (such as sprouting) in the bladder wall refractory to BoNT treatment. Unless so, it may be better to give a Table showing much evidence for resistance to BoNT treatment. Moreover, it would be better to give a Figure explaining the contents of sections 2.1 and 2.2 for the reader to easily understand a neuronal circuitry about bladder physiology. Throughout the text, there is inaccurate explanation about receptors such as vanilloid, purinergic and neurokinin receptors. There are several points that may be useful to amend this article, as follows:
The authors should mention references from which data given in Figures 2 and 3 are taken in the figures’ legends. Lines 15 and 56: were the data in Figures 2 and 3 taken from patients, refractory to BoNT treatment? If so, this fact should be mentioned in their figures’ legends. “Ach” should be “ACh” throughout the text. Line 80: ATP is adenosine triphosphate; please correct this spelling mistake. Lines 84 and 87: please expand “SNARE” and “CFTR”. Here, a short explanation of CFTR should be mentioned. Lines 84 and 172: the authors should explain “SV2” and “SNARE”. Please amend this point. Line 85: “chapter” should be “section”. Lines 177 and 182: not “VS2” but “SV2”? Please correct this point. Lines 204 and 205: does substance P activate both NK1 and NK2? Please reply to this question. Line 207: please expand TRPV1. Line 211: please explain P2X3 shortly. Line 226: does a purine mean ATP and adenosine? Please reply to this question. Lines 237-238: do vanilloid, muscarinic .. belong to tachykinin? Please correct this sentence. Do purinergic (both the metabotropic and ionotropic ones) mean P2Y and P2X? Please correct this point. Line 295: not “m” but “mm”? Please correct this spelling mistake. There appear to be more mistakes in English and scientific writing. This manuscript should be checked very carefully.